# EFFICIENT CALCULATION OF POLYNOMIAL FEATURES ON SPARSE MATRICES

**Nystrom, Andrew**
awnystrom@gmail.com [*][†]

Hughes, John
jfh@cs.brown.edu [†]

## ABSTRACT

We provide an algorithm for polynomial feature expansion that both operates on and produces a compressed sparse row matrix without any densification. For a vector of dimension $D$, density $d$, and degree $k$ the algorithm has time complexity $O(d^k D^k)$ where $k$ is the polynomial-feature order; this is an improvement by a factor $d^k$ over the standard method.

## 1 INTRODUCTION

Polynomial feature expansion has long been used in statistics to approximate nonlinear functions Gergonne (1974); Smith (1918). The compressed sparse row (CSR) matrix format is a widely-used data structure to hold design matrices for statistics and machine learning applications. However, polynomial expansions are typically not performed directly on sparse CSR matrices, nor on any sparse matrix format for that matter, without intermediate densification steps. This densification not only adds extra overhead, but wastefully computes combinations of features that have a product of zero, which are then discarded during conversion into a sparse format.

We provide an algorithm that allows CSR matrices to be the input of a polynomial feature expansion without any densification. The algorithm leverages the CSR format to only compute products of features that result in nonzero values. This exploits the sparsity of the data to achieve an improved time complexity of $O(d^k D^k)$ on each vector of the matrix where $k$ is the degree of the expansion, $D$ is the dimensionality, and $d$ is the density. The standard algorithm has time complexity $O(D^k)$. Since $0 \leq d \leq 1$, our algorithm is a significant improvement. While the algorithm we describe uses CSR matrices, it could be modified to operate on other sparse formats.

## 2 PRELIMINARIES

Matrices are denoted by uppercase bold letters thus: $\boldsymbol{A}$. The $i$the row of $\boldsymbol{A}$ is written $\boldsymbol{a}_i$. All vectors are written in bold, and $\boldsymbol{a}$, with no subscript, is a vector.

A compressed sparse row (CSR) matrix representation of an $r$-row matrix $\boldsymbol{A}$ consists of three vectors: $\boldsymbol{c}$, $\boldsymbol{d}$, and $\boldsymbol{p}$ and a single number: the number of columns of $\boldsymbol{A}$. The vectors $\boldsymbol{c}$ and $\boldsymbol{d}$ contain the same number of elements, and hold the column indices and data values, respectively, of all nonzero elements of $\boldsymbol{A}$. The vector $\boldsymbol{p}$ has $r$ entries. The values in $\boldsymbol{p}$ index both $\boldsymbol{c}$ and $\boldsymbol{d}$. The $i$th entry $\boldsymbol{p}_i$ of $\boldsymbol{p}$ tells where the data describing nonzero columns of $\boldsymbol{a}_i$ are within the other two vectors: $\boldsymbol{c}_{\boldsymbol{p}_i:\boldsymbol{p}_{i+1}}$ contain the column indices of those entries; $\boldsymbol{d}_{\boldsymbol{p}_i:\boldsymbol{p}_{i+1}}$ contain the entries themselves. Since only nonzero elements of each row are held, the overall number of columns of $\boldsymbol{A}$ must also be stored, since it cannot be derived from the other data.

Scalars, vectors, and matrices are often referenced with the superscript $k$. This is not to be interpreted as an exponent, but to indicate that it is the analogous aspect of that which procedes it, but in its polynomial expansion form. For example, $\boldsymbol{c}^2$ is the vector that holds columns for nonzero values in $\boldsymbol{A}$'s quadratic feature expansion CSR representation.

For simplicity in the presentation, we work with polynomial expansions of degree 2, but continue to use the exponent $k$ to show how the ideas apply in the general case.

---

[*]Now at Google
[†]The authors contributed equally important and fundamental aspects of this work.

We do provide an algorithm for third degree expansions, and derive the big-O time complexity of the general case.

We have also developed an algorithm for second and third degree interaction features (combinations without repetition), which can be found in the implementation.

# 3 MOTIVATION

In this section, we present a strawman algorithm for computing polynomial feature expansions on dense matrices. We then modify the algorithm slightly to operate on a CSR matrix, in order to expose its infeasibility in that context. We then show how the algorithm would be feasible with an added component, which we then derive in the following section.

## 3.1 DENSE EXPANSION ALGORITHM

A natural way to calculate polynomial features for a matrix $A$ is to walk down its rows and, for each row, take products of all $k$-combinations of elements. To determine in which column of $A_i^k$ products of elements in $A_i$ belong, a simple counter can be set to zero for each row of $A$ and incremented efter each polynomial feature is generated. This counter gives the column of $A_i^k$ into which each expansion feature belongs.

SECOND ORDER ($k = 2$) DENSE POLYNOMIAL EXPANSION ALGORITHM($A$)
1   $N = $ row count of $A$
2   $D = $ column count of $A$
3   $A^k = $ empty $N \times \binom{D}{2}$ matrix
4   **for** $i = 0$ **to** $N - 1$
5       $c_p = 0$
6       **for** $j_1 = 0$ **to** $D - 1$
7           **for** $j_2 = j_1$ **to** $D - 1$
8               $A_{ic_p}^k = A_{ij_1} \cdot A_{ij_2}$
9               $c_p = c_p + 1$

## 3.2 IMPERFECT CSR EXPANSION ALGORITHM

Now consider how this algorithm might be modified to accept a CSR matrix. Instead of walking directly down rows of $A$, we will walk down sections of $c$ and $d$ partitioned by $p$, and instead of inserting polynomial features into $A^k$, we will insert column numbers into $c^k$ and data elements into $d^k$.

INCOMPLETE SECOND ORDER ($k = 2$) CSR POLYNOMIAL EXPANSION ALGORITHM($\boldsymbol{A}$)

```
1    N = row count of A
2    pᵏ = vector of size N + 1
3    p₀ᵏ = 0
4    nnzᵏ = 0
5    for i = 0 to N − 1
6        i_start = pᵢ
7        i_stop = pᵢ₊₁
8        cᵢ = c_{i_start:i_stop}
9        nnzᵢᵏ = (|cᵢ| choose 2)
10       nnzᵏ = nnzᵏ + nnzᵢᵏ
11       pᵢ₊₁ᵏ = pᵢᵏ + nnzᵢᵏ

     // Build up the elements of pᵏ, cᵏ, and dᵏ
12   pᵏ = vector of size N + 1
13   cᵏ = vector of size nnzᵏ
14   dᵏ = vector of size nnzᵏ
15   n = 0
16   for i = 0 to N − 1
17       i_start = pᵢ
18       i_stop = pᵢ₊₁
19       cᵢ = c_{i_start:i_stop}
20       dᵢ = d_{i_start:i_stop}
21       for c₁ = 0 to |cᵢ| − 1
22           for c₂ = c₁ to |cᵢ| − 1
23               dₙᵏ = d_{c₀} · d_{c₁}
24               cₙᵏ =?
25               n = n + 1
```

The crux of the problem is at line 24. Given the arbitrary columns involved in a polynomial feature of $\boldsymbol{A}_i$, we need to determine the corresponding column of $\boldsymbol{A}_i^k$. We cannot simply reset a counter for each row as we did in the dense algorithm, because only columns corresponding to nonzero values are stored. Any time a column that would have held a zero value is implicitly skipped, the counter would err.

To develop a general algorithm, we require a mapping from columns of $\boldsymbol{A}$ to a column of $\boldsymbol{A}^k$. If there are $D$ columns of $\boldsymbol{A}$ and $\binom{D}{k}$ columns of $\boldsymbol{A}^k$, this can be accomplished by a bijective mapping of the following form:

$$(j_0, j_1, \ldots, j_{k-1}) \rightarrowtail p_{j_0 j_1 \ldots i_{k-1}} \in \{0, 1, \ldots, \binom{D}{k} - 1\} \tag{1}$$

such that $0 \leq j_0 \leq j_1 \leq \cdots \leq j_{k-1} < D$ where $(j_0, j_1, \ldots, j_{k-1})$ are elements of $\boldsymbol{c}$ and $p_{j_0 j_1 \ldots i_{k-1}}$ is an element of $\boldsymbol{c}^k$.

## 4    CONSTRUCTION OF MAPPING

Within this section, $i$, $j$, and $k$ denote column indices. For the second degree case, we seek a map from matrix indices $(i, j)$ (with $0 \leq i < j < D$ ) to numbers $f(i, j)$ with $0 \leq f(i, j) < \frac{D(D-1)}{2}$, one that follows the pattern indicated by

$$\begin{bmatrix} x & 0 & 1 & 3 \\ x & x & 2 & 4 \\ x & x & x & 5 \\ x & x & x & x \end{bmatrix} \tag{2}$$

where the entry in row $i$, column $j$, displays the value $f(i, j)$. We let $T_2(n) = \frac{1}{2}n(n + 1)$ be the $n$th triangular number; then in Equation 2, column $j$ (for $j > 0$) contains entries with $T_2(j − 1) \leq$

$e < T_2(j)$; the entry in the $i$th row is just $i + T_2(j-1)$. Thus we have $f(i,j) = i + T_2(j-1) = \frac{1}{2}(2i + j^2 - j)$. For instance, in column $j = 2$ in our example (the *third* column), the entry in row $i = 1$ is $i + T_2(j-1) = 1 + 1 = 2$.

With one-based indexing in both the domain and codomain, the formula above becomes $f_1(i,j) = \frac{1}{2}(2i + j^2 - 3j + 2)$.

For *polynomial* features, we seek a similar map $g$, one that also handles the case $i = j$. In this case, a similar analysis yields $g(i,j) = i + T_2(j) = \frac{1}{2}(2i + j^2 + j + 1)$.

To handle *three-way interactions*, we need to map triples of indices in a 3-index array to a flat list, and similarly for higher-order interactions. For this, we'll need the tetrahedral numbers $T_3(n) = \sum_{i=1}^{n} T_2(n) = \frac{1}{6}(n^3 + 3n^2 + 2n)$.

For three indices, $i, j, k$, with $0 \le i < j < k < D$, we have a similar recurrence. Calling the mapping $h$, we have

$$h(i,j,k) = i + T_2(j-1) + T_3(k-2); \tag{3}$$

if we define $T_1(i) = i$, then this has the very regular form

$$h(i,j,k) = T_1(i) + T_2(j-1) + T_3(k-2); \tag{4}$$

and from this the generalization to higher dimensions is straightforward. The formulas for "higher triangular numbers", i.e., those defined by

$$T_k(n) = \sum_{i=1}^{n} T_{k-1}(n) \tag{5}$$

for $k > 1$ can be determined inductively.

The explicit formula for 3-way interactions, with zero-based indexing, is

$$h(i,j,k) = 1 + (i-1) + \frac{(j-1)j}{2} + \tag{6}$$

$$\frac{(k-2)^3 + 3(k-2)^2 + 2(k-2)}{6}. \tag{7}$$

## 5 FINAL CSR EXPANSION ALGORITHM

With the mapping from columns of $\boldsymbol{A}$ to a column of $\boldsymbol{A}^k$, we can now write the final form of the innermost loop of the algorithm from 3.2. Let the mapping for $k = 2$ be denoted $h^2$. Then the innermost loop becomes:

```
for c_2 = c_1 to |c_i| - 1
    j_0 = c_{c_0}
    j_1 = c_{c_1}
    c_p = h^2(j_0, j_1)
    d_n^k = d_{c_0} · d_{c_1}
    c_n^k = c_p
    n = n + 1
```

The algorithm can be generalized to higher degrees by simply adding more nested loops, using higher order mappings, modifying the output dimensionality, and adjusting the counting of nonzero polynomial features in line 9.

## 6 TIME COMPLEXITY

### 6.1 ANALYTICAL

Calculating $k$-degree polynomial features via our method for a vector of dimensionality $D$ and density $d$ requires $\binom{dD}{k}$ (with repetition) products. The complexity of the algorithm, for fixed $k \ll$

$dD$, is therefore

$$O\left(\binom{dD + k - 1}{k}\right) = O\left(\frac{(dD + k - 1)!}{k!(dD - 1)!}\right) \tag{8}$$

$$= O\left(\frac{(dD + k - 1)(dD + k - 2)\dots(dD)}{k!}\right) \tag{9}$$

$$= O\left((dD + k - 1)(dD + k - 2)\dots(dD)\right) \text{ for } k \ll dD \tag{10}$$

$$= O\left(d^k D^k\right) \tag{11}$$

### 6.2 EMPIRICAL

To demonstrate how our algorithm scales with the density of a matrix, we compare it to the traditional polynomial expansion algorithm in the popular machine library scikit-learn Pedregosa et al. (2011) in the task of generating second degree polynomial expansions. Matrices of size $100 \times 5000$ were randomly generated with densities of $0.2, 0.4, 0.6, 0.8$, and $1.0$. Thirty matrices of each density were randomly generated, and the mean times (gray) of each algorithm were plotted. The red or blue width around the mean marks the third standard deviation from the mean. The time to densify the input to the standard algorithm was not counted.

The standard algorithm's runtime stays constant no matter the density of the matrix. This is because it does not avoid products that result in zero, but simply multiplies all second order combinations of features. Our algorithm scales quadratically with respect to the density. If the task were third degree expansions rather than second, the plot would show cubic scaling.

The fact that our algorithm is approximately $6.5$ times faster than the scikit-learn algorithm on $100 \times 5000$ matrices that are entirely dense is likely a language implementation difference. What matters is that the time of our algorithm increases quadratically with respect to the density in accordance with the big-O analysis.

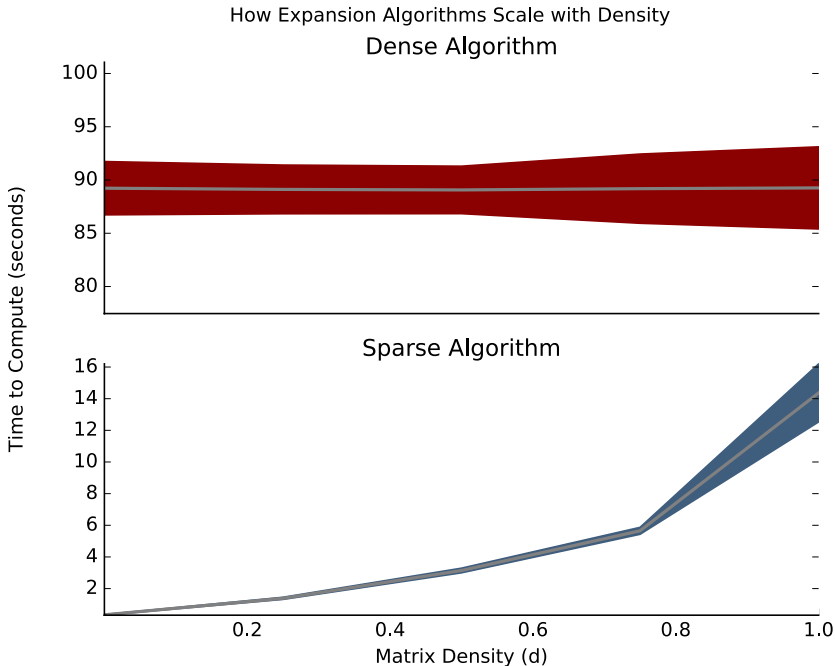

Figure 1: Our algorithm (bottom) scales with the density of a matrix, unlike the traditional polynomial feature expansion method (top). The task was a second degree expansion, which is why the time of our algorithm scales quadratically with the density.

## 7 CONCLUSION

We have developed an algorithm for performing polynomial feature expansions on CSR matrices that scales polynomially with respect to the density of the matrix. The areas within machine learning that this work touches are not en vogue, but they are workhorses of industry, and every improvement in core representations has an impact across a broad range of applications.

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
