# Peer review of "Efficient Calculation of Polynomial Features on Sparse Matrices"

_ICLR 2017 — rejected_

[Official Review · AnonReviewer3 · rating 3 · confidence 1 · 18 Dec 2016]
**This is not quite related with ICLR's field of interests**

The paper is beyond my expertise. I cannot give any solid review comments regarding the techniques that are better than an educated guess.

However, it seems to me that the topic is not very relevant to the focus of ICLR. Also the quality of writing requires improvement, especially literature review and experiment analysis.

[Official Review · AnonReviewer1 · rating 3 · confidence 3 · 19 Dec 2016]
**No Title**

This paper proposes an algorithm for polynomial feature expansion on CSR matrices, which reduces the time complexity of the standard method by a factor d^k where d is the density of the sparse matrix. The main contribution of this work is not significant enough. The experiments are incomplete and not convincing.

The background of the problem is not sufficiently introduced. There are only two references in the introduction part (overall only three papers are cited), which are from decades ago. Many more relevant papers should be cited from the recent literature.

The experiment part is very weak. This paper claims that the time complexity of their algorithm is O(d^k D^k), which is an improvement over standard method O(D^k) by a factor d^k. But in the experiments, when d=1, there is still a large gap (~14s vs. ~90s) between the proposed method and the standard one. The authors explain this as "likely a language implementation", which is not convincing. To fairly compare the two methods, of course you need to implement both in the same programming language and run experiments in the same environment. For higher degree feature expansion, there is no empirical experiments to show the advantage of the proposed method.

Some minor problems are listed below.
1) In Section 2, the notation "p_i:p_i+1" is not clearly defined.
2) In Section 3.1, typo: "efter" - "after"
3) All the algorithms in this paper are not titled. The input and output is not clearly listed.
4) In Figure 1, the meaning of the colored area is not described. Is it standard deviation or some quantile of the running time? How many runs of each algorithm are used to generate the ribbons? Many details of the experimental settings are missing.

[Official Review · AnonReviewer2 · rating 3 · confidence 3 · 21 Dec 2016]
**Interesting algorithm, but poor fit with ICLR**

The authors present here a new algorithm for the effective calculation of polynomial features on Sparse Matrices. The key idea is to use a proper mapping between matrices and their polynomial versions, in order to derive an effective CSR expansion algorithm. The authors analyse the time complexity in a convincing way with experiments.

Overall, the algorithm is definitely interesting, quite simple and nice, with many possible applications. The paper is however very superficial in terms of experiments, or applications of the proposed scheme. Most importantly, the fit with the main scope of ICLR is far from obvious with this work, that should probably re-submitted to better targets.

[Final Decision · Program Chairs · 06 Feb 2017]
**ICLR committee final decision**

The approach/problem seems interesting, and several reviewers commented on this. However, the experimental evaluation is quite preliminary and the paper would be helped a lot with a connection to a motivating application. All of the reviewers pointed out that the work is not written in the usual in the scope of ICLR papers, and putting these together at this time it makes sense to reject the paper.